# Seismic Performance of Drained Piles in Layered Soils

**DOI:** 10.3390/ma16175868

**Published:** 2023-08-27

**Authors:** Yaohui Yang, Gongfeng Xin, Yumin Chen, Armin W. Stuedlein, Chao Wang

**Affiliations:** 1Shandong Hi-Speed Group Innovation Research Institute, Jinan 250014, China; 2Key Laboratory of Ministry of Education for Geomechanics and Embankment Engineering, Hohai University, Nanjing 210098, China; 3College of Civil and Transportation Engineering, Hohai University, Nanjing 210098, China; 4School of Civil and Construction Engineering, Oregon State University, Corvallis, OR 97331, USA; 5School of Traffic & Transportation Engineering, Changsha University of Science & Technology, Changsha 410114, China

**Keywords:** drainage piles, shake table tests, liquefiable soils, excess pore pressure, discharge flow

## Abstract

The provision of drains to geotechnical elements subjected to strong ground motion can reduce the magnitude of shaking-induced excess pore pressure and the corresponding loss of soil stiffness and strength. A series of shaking table tests were conducted within layered soil models to investigate the effectiveness of drained piles to reduce the liquefaction hazard in and near pile-improved ground. The effect of the number of drains per pile and the orientation of the drains relative to the direction of shaking were evaluated in consideration of the volume of porewater discharged, the magnitude of excess pore pressure generated, and the amount of de-amplification in the ground’s motion. The following main conclusions can be drawn from this study. Single, isolated piles and a group of drained piles were tested in three series of shake table tests. Relative to conventional piles, the drained piles exhibited improved performance with regard to the generation and dissipation of excess pore pressure and stiffness of the surrounding soil, with increases in performance correlated with increases in the discharge capacity of the drained pile. The acceleration time histories observed within the pile-improved soil indicated a coupling of the rate and magnitude of porewater discharge, excess pore pressure generated, and de-amplification of strong ground motion. The amount of de-amplification reduced with increases in the number of drains per pile and corresponding reductions in excess pore pressure. The improved performance should prove helpful in the presence of sloping ground characterized with low-permeability soil layers that inhibit the dissipation of pore pressure and have demonstrated the significant potential for post-shaking slope deformation.

## 1. Introduction

The consequences of earthquake-induced soil liquefaction has produced a significant amount of damage to civil infrastructure, with the manifestation of excessive settlement, tilt, and lateral movement of buildings, bridges, lifelines, and waterfront structures [1,2,3,4,5] The 1964 Niigata and Good Friday, Anchorage, earthquakes provided engineers with the first modern opportunities to document the devastating effects of liquefaction. Hamada [6] describes observations of ruptured concrete piles discovered twenty years following the Niigata earthquake, with lateral deformations exceeding 0.6 m at their heads and clear indication of plastic hinging some 2 m below the pile connection. The numerous earthquakes since then have continued to form the basis for the evaluation of liquefaction susceptibility, triggering, and its consequences, and researchers have put significant effort into improving the understanding of soil–pile interaction in liquefying soil [7,8,9,10,11,12,13].

Numerous ground-improvement methodologies have been developed and refined to address the need to prevent damaging magnitudes of lateral deformation. Ground improvements include densification via dynamic compaction [14], vibro-compaction [15], drilled and driven displacement piles [16,17,18,19], and sand compaction piles [20]; reinforcement using vibro-replacement and drilled piles; seismic isolation using damping materials [21,22], deep-soil mixed columns and panels, and jet grout columns [23,24,25,26,27,28,29]; and drainage [30,31,32]. While some methods (e.g., displacement-type ground improvements) provide densification and reinforcement, vibro-compaction (i.e., stone columns) offers some efficiency by providing densification, reinforcement, and drainage, depending on the degree of mixing with native fines [25,26]. In the presence of significant silty fines, the installation of stone columns following improvement with pre-fabricated vertical drains (PVDs) can lead to improvements in the magnitude of densification [33,34]. However, the use of two or more different ground-improvement technologies can lead to significant costs associated with mobilization, scheduling, and coordination. Therefore, efforts to develop multi-function ground-improvement methodologies that can lead to further efficiencies in mitigating the consequences of liquefaction is of interest to the profession [18].

The inclusion of drainage within liquefying soils is one technology that continues to hold substantial merit for restraining large lateral deformations. Seed and Booker [35] proposed a methodology for incorporating drainage into the design of stone columns, which spurred significant research interest and investigation. The authors of [36,37,38,39,40,41,42] describe steel pipes and sheet piles fitted with drains to serve as a liquefaction countermeasure; shake table tests showed that the provision of drainage led to sufficient dissipation of excess pore pressure so as to prevent large deformations. Further recent developments have considered the range of drainage from the improvement in densification of silty sands with drained piles [18], the mitigation of liquefaction below embankments using plastic board PVDs [43], the direct in situ evaluation of the cyclic response of PVD-improved ground [44], and centrifuge testing [45]. When installed with deep foundations, drains have been shown to reduce the magnitude of excess pore pressure as well as the shaking-induced bending moments [46].

This paper demonstrates the performance of a new drained pile that combines the provision of vertical drains along the longitudinal axis of pre-cast piles. Shaking table tests were carried out on various model configurations to investigate the performance of drainage piles with a view of the discharge volumes of porewater, the generation and dissipation of shaking-induced excess pore pressure, and the de-amplification of acceleration. The experimental investigation focused on the orientation of drains relative to the direction of shaking, the number of drains per pile, and the response of single piles and groups of piles. The volume of porewater discharged, the excess pore pressures, and the acceleration response of the soil and piles were strongly affected by the number of drains and the orientation of the drain relative to the predominant direction of shaking. These results indicate that drained piles can serve to improve the seismic response of pile-improved ground and pile-supported structures without the use of separate construction equipment.

## 2. Drainage Piles: Concept and Prototypes

### 2.1. General Concept

The drainage pile evaluated in this paper combines the advantages of the flexural stiffness provided by a prefabricated (i.e., pre-cast) concrete pile with the ability to dissipate excess pore pressure via vertical drainage. Grooves are cast along the full length of the pile along its longitudinal axis, facilitating the placement of pre-fabricated vertical drains. The shape and size of the grooves can be modified to accommodate various drainage elements. For example, if cast in a circular arrangement, prefabricated, flexible steel-wire drain pipes can be readily installed to provide filtration and drainage. If the grooves are quadrate, rectangular PVDs can be used. The use of a geotextile filter fabric to cover the drain prevents clogging during installation and while in service. Owing to the small surface area of the drain along the pile relative to the remainder of the rough concrete surface, there is no significant impact on the resistance of axial loads through shaft resistance. Similar to the drainage piles tested by Stuedlein et al. [18], drainage of driving-induced excess pore pressure through contractive, silty sands can lead to increases in densification around the pile during pile installation, depending on the pile spacing, providing an additional benefit of the drainage pile. During shaking, the provision of drains with sufficient discharge capacity allows for the reduction in the loss of the mean effective stress surrounding the pile shaft and a corresponding reduction in lateral deformations arising from soil liquefaction. The first application of this kind of drainage pile was in Jiangyin, Jiangsu Province of China [47,48,49], using 23 m long, 0.3 m square precast concrete piles with a single rectangular drain running the length of the pile. The structure was supported on 203 of these piles distributed under the footprint area of 1100 m^2^.

### 2.2. Model-Scale Drainage Piles

The drainage piles evaluated in this study are shown in Figure 1. The overall model pile cross-section is 5 cm × 5 cm and is 60 cm long, and it is constructed of micro-concrete, which consists of cement, sand, and water using a ratio of 1:2:0.5, respectively. Considering the selected scale factor of 1:10 described below, these model piles represent 0.5 m square prototype piles, 6 m in length. Various drain configurations were investigated in the model tests, including a conventional pile, singly and doubly drained piles (i.e., with one and two drains, respectively), and a configuration with four drains (i.e., one drain on each side of the pile). The drain consisted of a rectangular plastic drainage board-type PVD 15 mm in width and 10 mm in depth and fitted with a geotextile to prevent clogging by the surrounding sand. The permeability coefficient of the geotextile was 0.162 cm/s.

### 2.3. Shaking Table Tests

Significant improvements in the understanding of the mechanisms of soil liquefaction and soil–pile interaction have been achieved using 1 g shake table testing [10,18,50,51,52,53]. The study described here was conducted using the shake table facility at Chongqing University, capable of simultaneous horizontal and vertical shaking with a maximum base excitation and frequency of 2 g and 50 Hz, respectively. Shaking-table tests were conducted within a laminar soil box with inner dimensions of 950 mm in length, 850 mm in width, and 600 mm in height. The selection of model component dimensions considered established 1 g scaling laws [54], as described below.

#### 2.3.1. Model Geometry and Materials

Given the potential for large deformations of liquefiable soil capped with an impermeable crust [55], the shake table tests were conducted using a two-layer system consisting of an upper clay layer overlying loose, liquefiable sand. Test specimens were constructed by pluviating 7^#^ silica sand (G_s_ = 2.64, D_50_ = 0.17 mm, ρd_,max_ = 1.65 g/cm^3^, ρd,_min_ = 1.34 g/cm^3^), of similar gradation to Toyoura sand (Figure 2), with water from a controlled height to produce uniform relative densities of 45% within the 40 cm thick liquefiable sand layer. Pluviation was paused intermittently to facilitate placement of various instruments, described subsequently, at the target locations. The piles were fixed to the base of the laminar box container such that the sand was pluviated around the piles. A 10 cm thick capping layer of clay was placed on top of the sand to prevent the vertical dissipation of excess pore pressure and complete the shake table specimen. The clay cap material was sourced from Southwest China, and is characterized by a specific gravity, G_s_, of 2.72; an in-place void ratio, e, equal to 0.91; a dry density, ρd, of 1.73 g/m^3^; liquid and plastic limits, ω_L_ and ω_p_, respectively, of 71 and 42; and hydraulic conductivity of k = 4.2 × 10^−7^ cm/s. Figure 2 presents the particle size distribution of clay cap material. Following pluviation of the sand, the clay cap was hand-placed on the sand layer and around the piles, allowing the PVDs to discharge porewater through the cap for collection, and trimmed to produce a thickness of 10 cm.

#### 2.3.2. Instrumentation

Instruments were deployed to observe the performance of the drainage pile-improved ground, including the shaking-induced excess pore pressure, acceleration, and the flow of water discharged from the drains. The pore pressure transducers (PPTs) used were characterized with a capacity of 20 kPa and accuracy of 0.1% full-scale (i.e., 0.02 kPa). Accelerometers were characterized with a range of ±10 g and sensitivity of 500 mV/g over the frequencies of interest. The PPTs and accelerometers were placed 20 cm from the base of the model container and 5 cm and 10 cm (i.e., one and two pile diameters) from the piles. In some shake tests, several piles were instrumented with accelerometers to capture the motion at the pile heads. The discharge flow from the PVDs was measured using flow transducers and pumps, as shown in Figure 3, and were arranged to capture and measure water exiting the top of the model pile without applying pump suction to water below the clay cap material.

#### 2.3.3. Test Configurations

Figure 4 presents the experimental configuration of each test specimen, with corresponding test and pile designations shown in Table 1. The first shake table test (designated T1) included one conventional pile (i.e., a pile with no drainage provision; designated T1ND) and three piles with a single drain orientated parallel (T1SD0), perpendicular (T1SD90), and 45° (T1SD45) to the direction of shaking (Figure 4). The piles were located 35 and 20 cm from the edges of the laminar box and placed to act as isolated piles, enforced using an impermeable membrane to prevent flow to adjacent piles. The second shake table test (designated T2) also evaluated four piles (at the same locations as T1) but considered piles with two drains (designated T2DD0 and T2DD90) and with four drains (designated T2QD090 and T2QD45), which allowed further investigation into the most efficient orientation of drainage piles. The third shake table test designated T3 was conducted on a 3 × 3 pile group spaced at three pile widths (i.e., 15 cm) from one another and with single drains oriented parallel to the direction of shaking (designated T3GSD0) in order to study the response of a group of free-headed drainage piles.

#### 2.3.4. Ground Motion and Scaling

Uniaxial shaking was accomplished with a 10 sec sinusoidal motion with 5 Hz frequency that increased to its peak amplitude of 0.2 g over a duration of two seconds, remained constant for 6 s, and then reduced to zero over a two-second duration (Figure 5). In order to distinctly analyse the results, sinusoidal motion was adopted in the test. This motion allows a smooth ramp-up and -down of the shaking table and has been found appropriate for liquefaction testing [13]. The selected scale factor for these model tests was 1:10; as such, the ground motion acceleration amplitude is 1:1, whereas the frequency scales to 1/n^0.5^ (Table 2; [56]). Thus, an equivalent experiment at full scale would experience ten seconds of shaking with peak amplitude of 0.2 g and frequency of 2.2 Hz.

## 3. Shake Table Results and Analysis

### 3.1. Comparison of Shaking-Induced Discharge Flow Volumes

The drainage of porewater during strong ground motion to relieve excess pore pressure represents the key advantage associated with the use of the drainage pile. Figure 6 presents the discharge flow time histories for shake table tests conducted with the drainage piles. Figure 6a presents the discharge time histories for isolated, single piles fitted with one drain oriented at 0°, 45°, and 90° (i.e., T1SD0, T1SD45, and T1SD90, respectively) from the direction of shaking. During shaking, the drained piles exhibit different rates of shaking-induced discharge: the pile with the drain oriented 90° from the direction of shaking produced the greatest volume of porewater, whereas pile T1SD0 produced the least volume. Following the end of shaking, the rate of discharge (or discharge flow) gradually slowed over the first 20 s to slow to a near-zero flow thereafter. The total volume of pore water removed from the liquefiable soil equalled 201 and 448 mL for the T1SD0 and T1SD90, respectively. The isolated pile with a single drain oriented at 45° performed nearly the same as T1SD90, yielding a total porewater discharge volume of 393 mL (approximately 88% of T1SD90).

Figure 6b presents the discharge flow time histories for the isolated, single piles with two and four drains observed during the second shake table test (T2 series; Table 1). In the case of two drainage paths, the piles with two drains oriented parallel to the direction of shaking (i.e., T2DD0) produced greater discharge volumes than the comparable, singly-drained pile (T1SD0), with a total discharge volume of 333 mL. However, pile T2DD90 with drains oriented perpendicular to the direction of shaking removed nearly twice the porewater volume as T2DD0 at the end of shaking (i.e., 620 mL). Piles with four drainage paths oriented at 45° and 90° to the direction of shaking, designated T2QD45 and T2QD90, discharged 513 and 671 mL of porewater at the end of shaking. Thus, it appears that piles with two drains oriented perpendicular to the direction of shaking exhibit greater efficiency than piles with four drains if the orientation of the drains is rotated toward the predominant direction of shaking. However, given that earthquake motions exhibit significant variations in direction, arbitrary directionality seems to be best mitigated through the use of four drains per pile.

Figure 6c presents the discharge flow time histories for drained piles set within a pile group evaluated in the third shake table test (i.e., T3, Table 1). A single drain was fitted to each pile in this test series and all piles were oriented at 90° or perpendicular to the direction of shaking. The shaking-induced flow was monitored for four of the nine piles, located at the centre and one corner of the pile group, and a side pile leading the centre pile and one trailing the corner piles. The instrumented piles exhibited similar magnitudes of discharge flow during shaking; however, following shaking, the centre, corner, and the second side pile exhibited a greater amount of discharge flow, producing discharge volumes of 483, 445, and 408 mL, respectively. Side pile 1, trailing the corner piles, produced the smallest magnitude of post-shaking discharge volume of 245 mL.

Figure 7 compares the maximum discharge volume of isolated, single piles with one, two, and four drains and with drain orientations relative to the direction of shaking. Doubling the drainage capacity from one to two drains results in a significant increase in discharge volume for the orientations investigated. While doubling the drainage capacity from two to four drains results in further increases in discharge volume, the incremental increase in volume is smaller than that when doubling the capacity from one to two drains. This suggests that there may be limited benefit in providing four drainage paths to a drained pile; however, the predominant direction of shaking is usually not known with significant accuracy as noted earlier.

### 3.2. Comparison of Excess Pore Pressure Generation and Dissipation

Drained piles have the potential to mitigate the consequences of liquefaction through the accelerated dissipation of excess pore pressure during and immediately following strong ground motion. Figure 8 demonstrates the rate of excess pore pressure generation and dissipation during the shake table tests of the single, isolated piles in terms of the excess pore pressure ratio, r_u_, along with the distribution of PPTs for each pile. PPTs located 180° from the drains (i.e., P3 and P6 for T1SD90 and T1SD45, respectively) consistently produced the largest excess pore pressures, with peak r_u_ of about 0.88 (n. b., P9 did not function during the test, but likely would have produced a similar r_u_ response). On the other hand, PPTs directly in front of the drains consistently exhibited the smallest generation of excess pore pressure, with peak r_u_ ranging from 0.53 to 0.58. For these PPTs, the lowest peak r_u_ (i.e., 0.53) was observed for the drain arranged perpendicular to shaking (T1SD90; Figure 8a), whereas the highest peak r_u_ (i.e., 0.58) was generated for the case with the single drain oriented perpendicular to the direction of shaking (T1SD0; Figure 8c). Excess pore pressures generated along the sides of the pile and observed using PPTs P2, P5, and P8 ranged from 0.60 to 0.64; this indicates, along with the PPTs, that the drains produce an azimuthal variation in r_u_ and corresponding hydraulic gradient driving flow from the region away from the drain towards the drain. This implies that a preferred orientation of singly drained piles placed in sloping ground may exist in order to restrain ratchet-type movements downslope. Furthermore, the excess pore pressures observed for most of the PPTs dissipated to near-zero within 10 s following shaking; this represents a significant advantage of slopes with crust caps that can retard the dissipation of migrating porewater and lead to delayed failure [54,55].

In comparison, the single, isolated, conventional pile (T1ND, Figure 8d) tested to compare the r_u_ response in the absence of a drain produced significantly larger peak r_u_ values equal to 1.0 at all locations. For this pile, the duration to full liquefaction varied with location: P10 and P12, placed in front and behind the pile and parallel to the ground motion, exhibited liquefaction near-simultaneously at about 1.0 s into the ground motion, whereas P11, placed along the side of the pile in the direction perpendicular to the direction of shaking (relative to the pile), indicated the onset of liquefaction at 7.5 s of shaking. The rate of excess pore pressure dissipation following shaking observed in all PPTs was significantly lower than for the case of the drained piles. In general, the presence of a single drain on the model piles significantly reduced the rate and magnitude of excess pore pressure generated as compared to the conventional pile.

Figure 9 presents the rate of excess pore pressure generation and dissipation during the second shake table test series with piles fitted with two and four drains along with the distribution of PPTs for each pile. In general, the near-pile r_u_ time histories (observed using P1 and P3, P5 and P7, P9 and P11, and P13 and P15), indicate slightly lower magnitudes than the singly drained piles (compare to Figure 8). Comparison of T2QD090 (Figure 9a) to T2DD0 and T2DD90 (Figure 9c,d) indicate that the use of four drains results in reduced excess pore pressures, particularly in the region further away from the piles. The largest magnitude of excess pore pressure was observed using P12 for the doubly drained pile T2DD0, with peak r_u_ of about 0.86. The remainder of the PPTs exhibited peak r_u_ values of 0.59 or smaller: the average peak r_u_ for piles with two drains was 0.58, whereas the average maximum r_u_ for piles with four drains was 0.46. Thus, use of a greater number of drains will produce a greater volume of stabilized soil surrounding the pile, resulting in smaller permanent displacements if installed in sloping ground.

The excess pore pressure ratio time histories for the T3 test series of a group of singly-drained piles is shown in Figure 10. Comparison of the pore pressure response of the group of singly drained piles to single, isolated singly drained piles in Figure 8 indicates that the pile group serves to drain a larger volume of soil. The pore pressure response within or near the pile group (PPTs P1 through P8) exhibited an average peak r_u_ of 0.5, comparable to the case of single piles with four drainage paths. This indicates that singly-drained piles may be more cost-effective than multi-drained piles when installed in groups. Away from the pile group, the peak r_u_ ranged from 0.68 to 1.00 (PPTs P9 through P12), with largest excess pore pressure response for the PPTs furthest from the corner pile (i.e., P12) and characterized by the longest drainage path.

Comparison of the three test series indicates that the volume of porewater discharged varied with the number of drains per pile, the orientation of the drains relative to the direction of shaking, and the position of a drained pile within a group (Figure 6, Figure 7, Figure 8, Figure 9 and Figure 10). The presence of a drain serves to reduce r_u_ in the liquefiable soil as a function of the radial distance from the drain, as shown in Figure 11a for the T1 test series. Here, the excess pore pressure averaged over the period of the ground motion with constant amplitude (i.e., strong shaking from 2 to 8 s) is plotted with radial distance, indicating a sharp reduction in r_u_ to a near-constant magnitude as the proximity to the drain increases. The excess pore pressure field during shaking implied by Figure 11a does not appear significantly sensitive to the drain orientation; however, the total volume of porewater discharged (during and following shaking) was shown to vary with drain orientation relative to the direction of shaking (Figure 7). Figure 11b presents the variation in porewater volume discharged at time t = 10 s (at the end of shaking) and 40 s (representing total porewater discharged) with the average hydraulic gradient computed using the PPTs nearest and farthest from the drain for the T1 and T2 test series and during the period of strong, constant shaking. The hydraulic gradients increase with the decrease in the number of drains per pile and with increasing orientation away from the direction of shaking. Furthermore, at the end of shaking (t = 10 s), the effect of the average hydraulic gradient on the discharge volume increases with decreasing number of drains per pile. However, as time elapses following the end of shaking and excess pore pressures dissipate, the effect of increasing drain orientation away from the direction of shaking on discharge volume increases. These results suggest that the average hydraulic gradient developed during shaking controls the magnitude of porewater discharged until sufficient discharge capacity is provided (by the increased number of drains per pile), whereupon the volume of discharged porewater becomes controlled by the hydraulic conductivity of the soil. Head losses associated with porewater entering the drain increase with the increasing discharge velocities [57], hence increasing hydraulic gradients. These test results suggest that the ability of drained piles to quickly reduce excess pore pressures during and following shaking is more strongly controlled by the hydraulic gradient when discharge capacity is small (i.e., singly drained piles) and more strongly controlled by the orientation of the drained pile when discharge capacity is large.

### 3.3. Comparison of Acceleration Time Histories

The acceleration time histories observed in the soil adjacent to the piles, presented in Figure 12, demonstrated the role of drain orientation on the propagation of seismic energy through the soil–pile model. Note that the accelerometers were placed in different physical locations for each drained pile, but along the same orientation relative to the front and back of a given drained pile; therefore, some differences in the response may be attributed to the spatial distribution of shaking within the box and soil–pile–soil interaction relative to the direction of shaking. Pile T1SD90 (Figure 12a) exhibited the highest magnitude of sustained acceleration, as well as the largest peak horizontal acceleration (PHA) of 2.04 m/s^2^, as compared to T1SD45 (Figure 12b) and T1SD0 (Figure 12c). The acceleration time histories for T1SD90 indicate that the magnitude of shaking varies with distance from the drain, with higher magnitudes of acceleration corresponding to the smaller distances. This observation holds for drained piles oriented 45 and 0° from the direction of shaking.

Excess pore pressures observed for the single, isolated drained piles reduced with increasing proximity to the drain. Thus, the reduction in excess pore pressure is associated with increased soil stiffness, allowing for greater propagation of seismic energy through the soil mass at that location. This may be confirmed in Figure 12d, which shows that the full liquefaction produced for pile T1ND, with no provision for drainage, resulted in significant de-amplification of the input ground motion. The T1 test series also shows that as the volume of porewater discharged from the drains increases (Figure 6a and Figure 7), the magnitude of peak and sustained acceleration decrease. Thus, the direction of shaking relative to the drain serves to reduce the liquefaction hazard through a reduction in acceleration, which appears to lead to a reduction in excess pore pressure and increase discharge volume.

Figure 13 provides the acceleration time histories for the pile group observed in the soil adjacent to and top of selected piles (refer to Figure 4). Similar to the T1 series, the accelerometers were placed 10 cm above the base of the model and indicate the attenuation of the input ground motion to varying extents. In general, the comparison of Figure 13 and Figure 10 shows that accelerations measured closer to the centre of the pile group exhibited the least amount of de-amplification and was associated with the least magnitude of excess pore pressure. The acceleration time histories observed using A1 through A4, placed in or near the pile group, indicated similar responses and acceleration magnitudes. Acceleration magnitudes increased until the excess pore pressures reached relatively stable plateaus (at approximately 2 s of shaking; Figure 10), and then reduced with continuing shaking followed by a gradual rise (at about 6 s) as the drains began to discharge porewater (Figure 6c). The PHA for accelerometers A1 though A4 were observed towards the end of shaking and ranged from 1.78 to 1.95 m/s^2^. Accelerations measured furthest from the drained pile group (i.e., A5) produced the greatest reduction in magnitude, which was associated with greatest r_u_, as shown in Figure 10d.

Acceleration time histories measured at the top of the selected drained piles were largely similar those measured immediately adjacent to a given pile, indicating negligible damping associated with soil–pile–soil interaction. This may have resulted from the use of free-headed, unloaded piles in the pile group appropriate for mitigation of lateral spreading in the free field [13]. Pile groups constructed with a relatively rigid pile cap and subjected to inertial loading would likely have exhibited differential motion due to soil–pile–soil interactions (e.g., gapping and drag) that would likely result in a different response. The acceleration time histories varied slightly with pile position in the group: the maximum acceleration of a pile top was observed in the centre of the group, whereas the minimum acceleration was observed at Side Pile 1. These observations show that the discharge capacity of drained piles control the magnitude of excess pore pressures and de-amplification, and therefore, the strength and stiffness of the pile-improved ground, and that the interaction between discharge capacity, excess pore pressure, and acceleration can be inferred from accepted soil dynamics principles.

## 4. Conclusions

A series of shake table tests were conducted to investigate the effectiveness of drained piles in reducing the liquefaction hazard in and near pile-improved ground within a layered subsurface profile. Various configurations were evaluated, included piles with differing numbers of drains and orientations relative to the direction of shaking. The response of the drained pile-improved soil was evaluated using the discharged volume of porewater, the generation and dissipation of excess pore pressure, and the attenuation of acceleration. The following main conclusions can be drawn from this study:The volume of porewater discharged was dependent on the orientation of the drains relative to the predominant direction of shaking and the number of drains per pile. The efficiency of the drains increased as the orientation of the drains increased from 0° to 90° relative to the direction of shaking. The provision of two drains per pile resulted in a significant increase in the volume of porewater discharged as compared to the singly drained pile, but the incremental increase in discharge volume reduced when doubling the drains again to four per pile.Drained piles demonstrated the ability to significantly reduce the magnitude of shaking-induced excess pore pressures in proximity to the drain. The excess pore pressure ratio reduced slightly with an increase in the number of drains per pile; however, the extent of soil with reduced excess pressure increased with the increase in the discharge capacity or number of drains. The orientation of the drain relative to the direction of shaking also influenced the magnitude of excess pore pressure.The provision of drains to the model piles resulted in a sharper reduction in post-shaking excess pore pressure, a critical feature for layered soils that include a low-permeability crust, which can inhibit the dissipation of pore pressure and which may result in large post-shaking deformations when the crust is accompanied by sloping ground conditions.The amount of porewater discharged and excess pore pressure generated within and near the group of drained piles depended on the position of the pile within the group; however, the soil within the group exhibited relatively low variation in the distribution of excess pore pressure.The acceleration time histories observed within the pile-improved soil indicated a coupling of the rate and magnitude of porewater discharge, excess pore pressure generated, and de-amplification of strong ground motion. The amount of de-amplification reduced with increases in the number of drains per pile and corresponding reductions in excess pore pressure. Therefore, removal of excess pore pressure-driven porewater maintains the integrity of the ground motion due to prevention of stiffness degradation associated with liquefaction.

## Figures and Tables

**Figure 1 materials-16-05868-f001:**
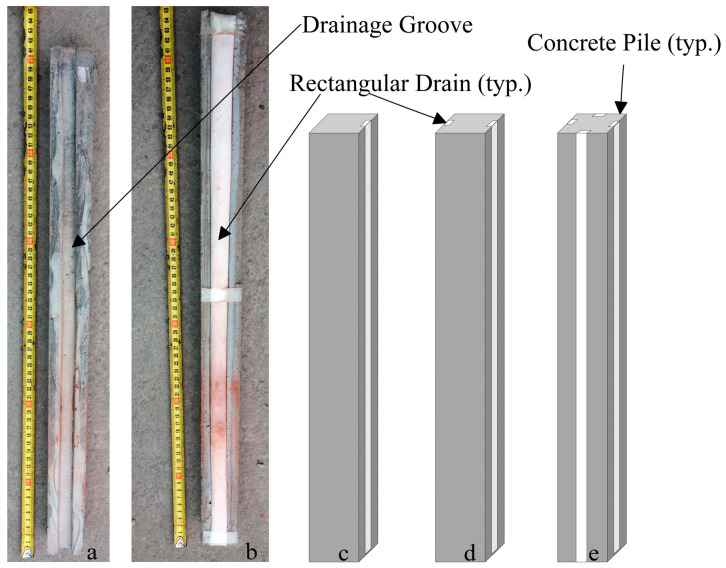
Drainage pile geometry investigated here: (**a**) model micro-concrete pile showing groove for drain, (**b**) model pile fitted with drain, schematic indicating typical square concrete pile with drain, (**c**) single drain, (**d**) two drains, and (**e**) four drains.

**Figure 2 materials-16-05868-f002:**
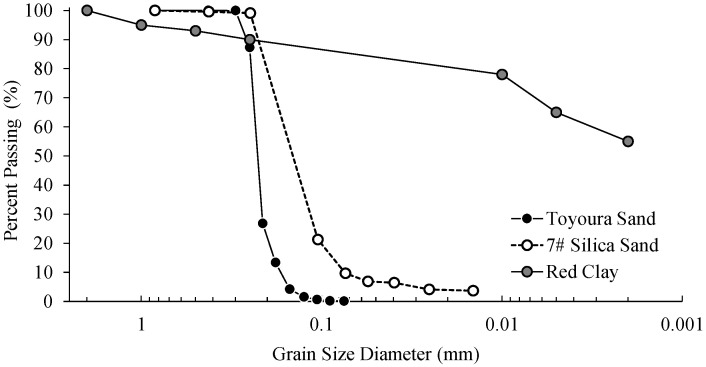
Particle size distributions of materials used in this study and comparison of 7# silica sand to Toyoura sand.

**Figure 3 materials-16-05868-f003:**
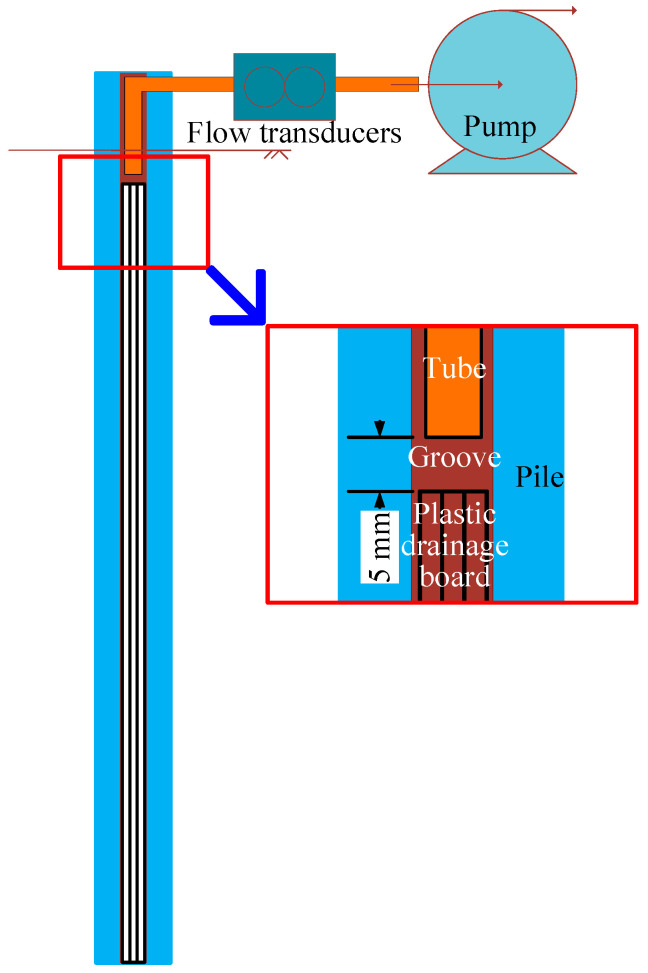
Schematic indicating provisions for measuring the volume of porewater discharged.

**Figure 4 materials-16-05868-f004:**
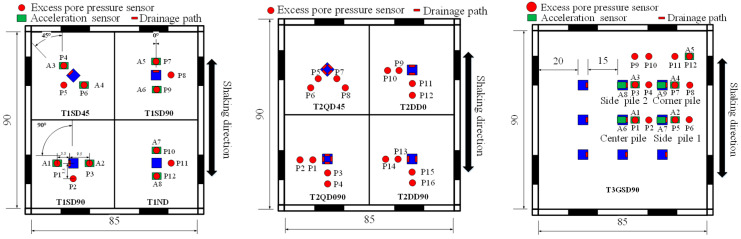
Plan, elevation, and instrumentation arrays used for shake table tests: (**a**) test series T1 with single piles, including T1ND, T1SD0, T1SD90, and T1SD45; and the blue square is flow transducer; the white line is impermeable foam board; the yellow line is tube; and (**b**) test series T2 with single piles with T2DD0, T2DD90, T2QD090, and T2QD45; and (**c**) test series T3 with a group of piles fitted with a single drain, T3GSD90 (Unit: cm).

**Figure 5 materials-16-05868-f005:**
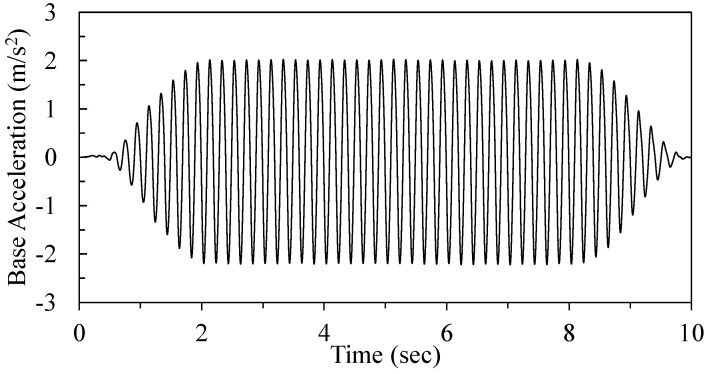
Acceleration time history applied by shake table for each experiment.

**Figure 6 materials-16-05868-f006:**
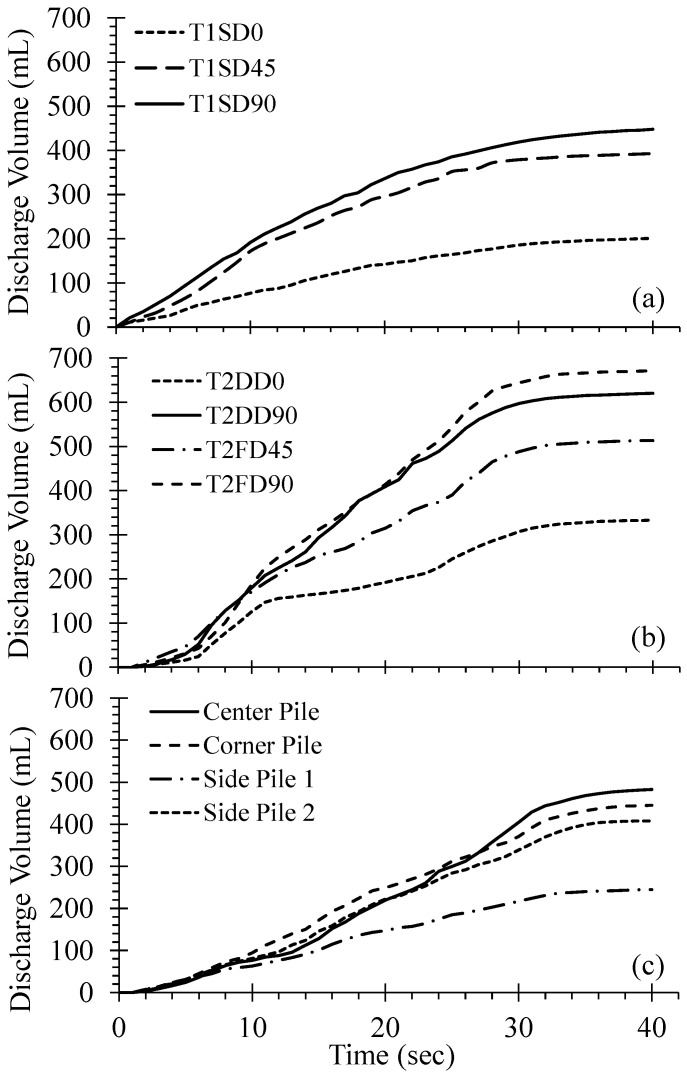
Comparison of discharge volume time histories for (**a**) test series T1 on single piles with one drain per pile, (**b**) test series T2 on single piles with two and four drains per pile, and (**c**) test series T3 on a group of singly-drained piles.

**Figure 7 materials-16-05868-f007:**
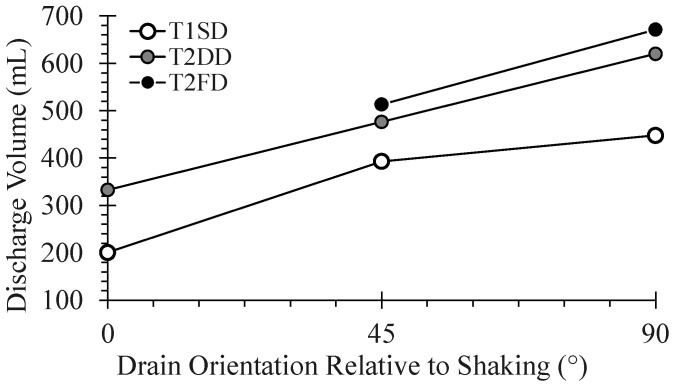
Comparison of discharge volumes observed for single piles with various number of drains and drain orientations relative to shaking.

**Figure 8 materials-16-05868-f008:**
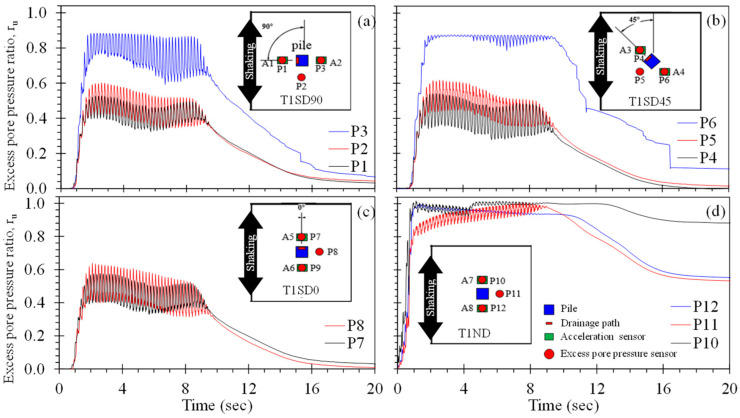
Excess pore pressure time histories for test series T1: (**a**) pile T1SD90, (**b**) pile T1SD45, (**c**) pile T1SD0, and (**d**) conventional pile T1ND.

**Figure 9 materials-16-05868-f009:**
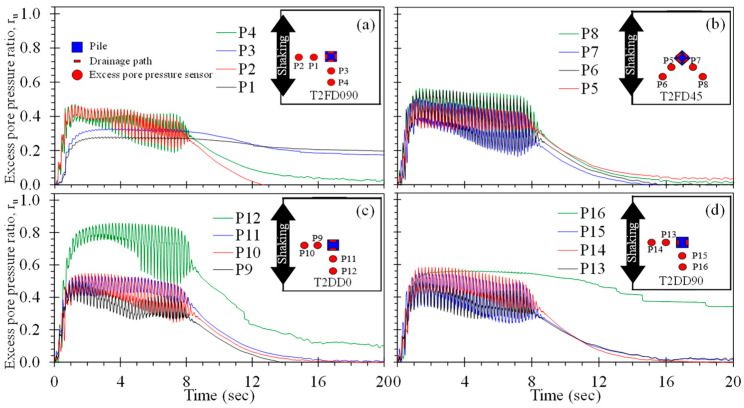
Excess pore pressure time histories for test series T2: (**a**) pile T2QD090, (**b**) pile T2QD45, (**c**) pile T2DD0, and (**d**) T2DD90.

**Figure 10 materials-16-05868-f010:**
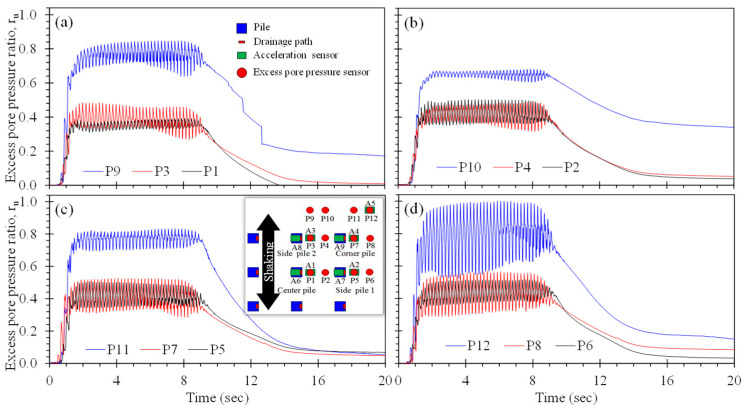
Excess pore pressure time histories for test series T3: (**a**) pore pressures observed along the centre of the pile group, (**b**) pore pressures observed between the centre and outer column of piles, (**c**) pore pressures observed immediately adjacent to the outer column of piles, and (**d**) pore pressures observed away from the outer column of piles.

**Figure 11 materials-16-05868-f011:**
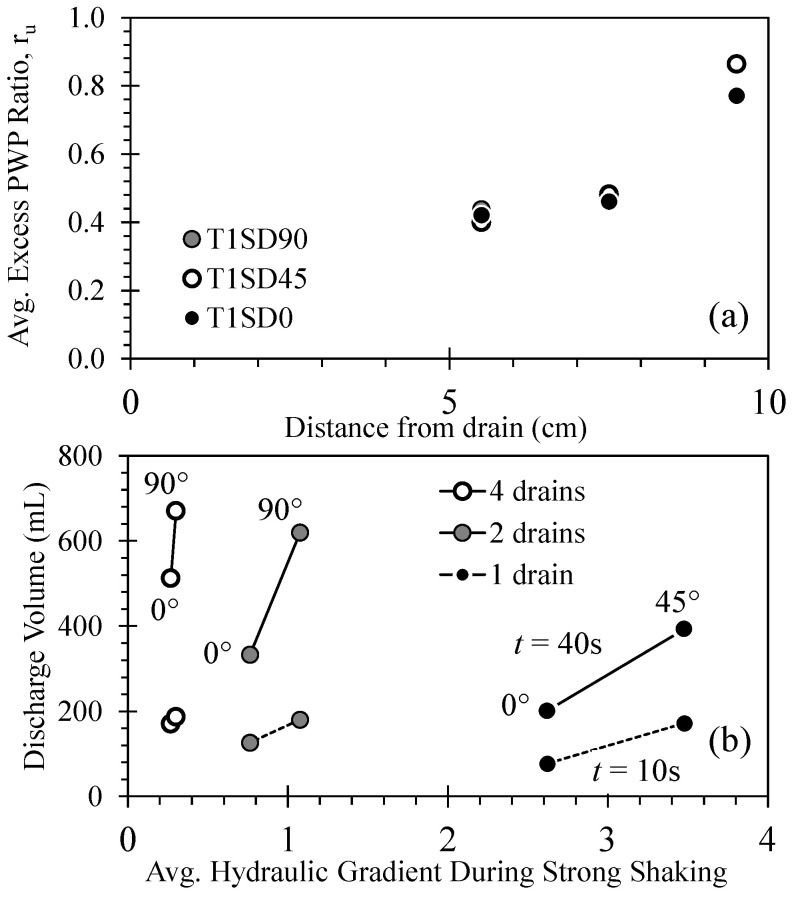
Summary of excess pore pressure response and volume of porewater discharged for test series T1 and T2: (**a**) variation in average excess pore pressure during strong shaking with radial distance from the drain and (**b**) variation in discharge volume at time t = 10 and 40 s with average hydraulic gradient during strong shaking.

**Figure 12 materials-16-05868-f012:**
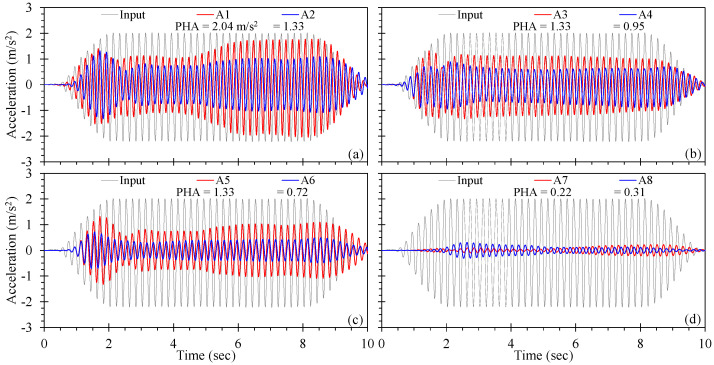
Acceleration time histories for test series T1: (**a**) pile T1SD90, (**b**) pile T1SD45, (**c**) pile T1SD0, and (**d**) conventional pile T1ND.

**Figure 13 materials-16-05868-f013:**
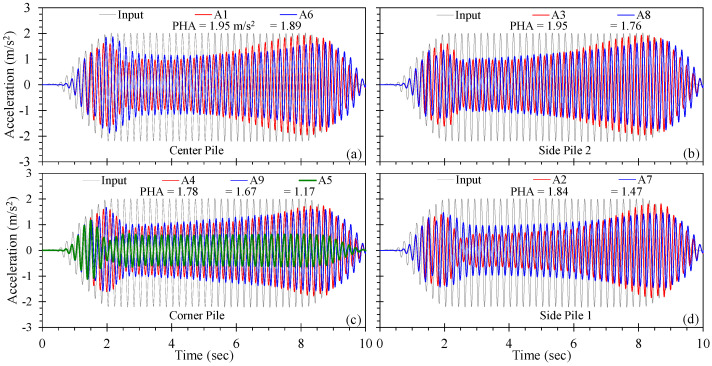
Acceleration time histories for test series T3: (**a**) adjacent to and on top of the centre pile, (**b**) adjacent to and on top of the second side pile, (**c**) adjacent to and on top of the corner pile, and (**d**) adjacent to and on top of the first side pile.

**Table 1 materials-16-05868-t001:** Shake table test configurations.

Shake Table Test Number	Pile Designation	Pile Configuration	Drainage Configuration	Drainage Area (cm^2^)	Orientation of Drain with Shaking Direction (°)
T1	T1ND	Single pile	No drain	0	N/a
T1	T1SD0	Single pile	Single drain	90	0
T1	T1SD90	Single pile	Single drain	90	90
T1	T1SD45	Single pile	Single drain	90	45
T2	T2DD0	Single pile	Double drain	180	0
T2	T2DD90	Single pile	Double drain	180	90
T2	T2QD090	Single pile	Quad drain	360	0, 90
T2	T2QD45	Single pile	Quad drain	360	±45
T3	T3GSD90	Group of piles	Single drain	810	90

**Table 2 materials-16-05868-t002:** Similitude laws for 1 g shaking table tests (after Iai, [56]).

Items	Model	Prototype
Scaling factor	1	n
Length	1/n	1
Density	1	1
Displacement	1/n	1
Stress	1	1
Frequency	1/n^0.5^	1
Acceleration	1	1

## Data Availability

Data available on request due to restrictions e.g., privacy or ethical. The data presented in this study are available on request from the corresponding author. The data are not publicly available because they contain proprietary information or trade secrets.

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
