# Peer review of "Seismic Performance of Drained Piles in Layered Soils"

_materials, 2023, doi:10.3390/ma16175868_

Round 1
Reviewer 1 Report
In the present article, the authors investigate the performance of the newly drained pile. The shaking table tests have been performed to investigate the performance of the drainage piles. The study is interesting and may be useful for the readers of the scientific society. Please find below review comments which may help the authors to improve the quality of the manuscript.
1. A grammatical error in line no. 13.
2. Typo error in Figure 1 caption.
3. The piles used for the tests are square. What would be the results if the piles are circular in geometry?
4. What would be the effect on the bearing capacity of the pile due to the drainage groove? What cross-sectional area is to be considered for calculating the strength at that particular section?
5. State a practical example of the micro concrete pile with drainage groove used?
6. The picture clarity in Figure 4 may be improved.
7. The authors should validate their results with analytical solutions previously published in the literature or three-dimensional numerical analysis.
8. The article is to be checked thoroughly for typo errors and grammatical mistakes if any.
9. The references are to be checked thoroughly.
It can be improved.
Reviewer 2 Report
The paper's objective is valuable, and the conducted experimental tests are commendable. However, the analysis of the experimental data lacks generalization, which could enhance the paper's overall contribution.
A crucial point to consider is whether the authors conducted a reference zero test without drainage. Including this test is essential to estimate the seismic demand reduction achieved by drainage. It's worth emphasizing that such a measure doesn't impact the pile's capacity; rather, it affects only the seismic demand. The paper could benefit significantly by addressing this aspect more comprehensively. Specifically, the authors should quantify the reduction in seismic demand achieved through drainage and discuss its implications. Moreover, exploring the possibility of predicting these results using classical geotechnical approaches, such as effective stress theory, would greatly enhance the paper's theoretical foundation.
On the presentation side, there is room for improvement. The quality of the figures needs attention, particularly the font size. The figures should align with the main text's font size, providing a visually consistent experience for readers. Additionally, enhancing the figures' aesthetics and informativeness would improve their overall impact.
Furthermore, the reference list could be expanded to encompass a broader scope of attempts aimed at improving the seismic performance of pile systems. Here are a couple of relevant references that could be included:
-
Aloisio, A., Contento, A., Xue, J., Fu, R., Fragiacomo, M., & Briseghella, B. (2023). Probabilistic formulation for the q-factor of piles with damping pre-hole. Bulletin of Earthquake Engineering, 21(8), 3749-3775.
-
Panah, A. K., & Khoshay, A. H. (2015). A new seismic isolation system: sleeved-pile with soil-rubber mixture. International Journal of Civil Engineering, 13(2), 124-132.
Round 2
Reviewer 1 Report
The authors have incorporated all the review comments. The article may be accepted.
It may be improved.
Reviewer 2 Report
Accept